# Psychometric evaluation of Korean version of COVID-19 fear scale (K-FS-8): A population based cross-sectional study

**Jung Jae Lee, Hye Ri Choi** [ORCID]‡**, Edmond Pui-Hang Choi, Mu-Hsing Ho, Daniel Y. T. Fong** [ORCID]*, **Kris Yuet Wan Lok, Mandy Ho, Chia-Chin Lin**

School of Nursing, The University of Hong Kong, Pok Fu Lam, Hong Kong SAR

‡ HRC are contribution as the first author on this work.
* dytfong@hku.hk

**Data Availability Statement:** Data are available on reasonable request due to no approval from the IRB obtained. Enquiries: Mr. Chris YIP (Tel:+852 2255 4086; E-mail: yipcf@ha.org.hk).

## Abstract

COVID-19-related fear negatively affects the public's psychological well-being and health behaviours. Although psychological distress including depression and anxiety under COVID-19 is well-established in literature, research scarcely evaluated the fear of COVID-19 with a large sample using validated scale. This study aimed to validate a Korean version of fear scale(K-FS-8) using an existing fear scale(Breast Cancer Fear Scale; 8 items) and to measure the fear of COVID-19 in South Korea. A cross-sectional online survey was conducted with 2235 Korean adults from August to September 2020. The Breast Cancer Fear Scale was translated from English into Korean using forward-backward translation, and then face validity was assessed. Patient Health Questionnaire-4 and Primary Care Post-Traumatic Stress Disorder Screen for DSM-5 were used for assessing convergent validity of K-FS-8, and item response theory analysis was also conducted to further validate the K-FS-8. This study confirmed the validity and reliability of the K-FS-8. The validity of the scale was confirmed by convergent validity, known-group validity and item response theory analysis, and internal consistency was also examined(Cronbach's α coefficient = 0.92). This study also identified that 84.6% participants had high COVID-19 fear; whilst 26.3%, 23.2% and 13.4% participants had high risk of post-traumatic stress disorder, depressive and anxiety symptoms, respectively. The K-FS-8 showed the acceptability measuring the fear of COVID-19 in the Korean population. The K-FS-8 can be applied to screen for fear of COVID-19 and related major public health crises identifying individuals with high levels of fear in primary care settings who will benefit from psychological support.

## 1. Introduction

Since the novel coronavirus disease 2019(COVID-19) emerged in 2019, as of July 2022, COVID-19 caused over 554 million confirmed cases and over 6.35 million deaths worldwide [1]. The World Health Organization reported that psychological distress among the public increased by 25% during the pandemic [2]. More specifically, the global prevalence of anxiety and depression had sharply increased by 25.6% and 27.6%, respectively, compared with the

**Funding:** The authors received no specific funding for this work.

**Competing interests:** The authors have declared that no competing interests exist.

prevalence before the pandemic [3]. Particularly, studies [2, 4, 5] reported that the social distancing measures to control the wide and rapid spread of COVID-19 precipitated psychological distress including the symptoms of depression, anxiety, post-traumatic stress syndrome (PTSD) and fear due to the life changes and decreased social contacts. The psychological distress with and without COVID-19 can cause clinically significant mental health disorders and has a negative impact on physical health [4–7]. Studies reported that the psychological distress was also associated with poor COVID-19 coping strategies and preventative health behaviours, resulting in a high risk of COVID-19 infection [8–10].

Fear, a type of psychological distress, indicates the typical role as a double-edged sword in health behaviours under the COVID-19 pandemic. Fear is defined as an individual's '*response to perceived threat that is consciously recognised as a danger*' [11, p. 244], and the threat of fear is supposed to be identifiable which therefore includes infectious diseases such as COVID-19 [12]. Whilst fear can play an important role in coping strategies for COVID-19 such as following the social distancing rules, wearing masks and washing hands [13], a higher level of fear was associated with detrimental health behaviours such as increased cigarette use and alcohol drinking [5]. In addition, fear of COVID-19 can cause the deterioration of psychological symptoms of the vulnerable population with underlying psychological issues [14].

In South Korea, over 18 million confirmed cases and 24,680 deaths were reported as of 13th July 2022 [15]. The social distancing rule was established by the South Korean government in late February 2020 [16]. Accordingly, a growing number of psychological distresses such as depression and anxiety have been reported [17, 18]. A survey study conducted at the beginning of the COVID-19 pandemic 2020 reported that 45% of their respondents had clinical levels of depression, anxiety or stress [17]. Another survey study addressed that 30.7% of participants were at risk of depression, and 22.6% were at the risk of anxiety [18]. However, fear of COVID-19 in South Korea has not been investigated using a large sample. Therefore, this study aimed to measure the COVID-19 related fear with a large sample of Korean population during COVID-19 pandemic, by conducting a psychometric evaluation of the Korean version of fear scale. This study adopted the Breast Cancer Fear Scale (BCFS; 8 items) [19] for the psychometric evaluation. The BCFS was developed on the robust theoretical frameworks–Extended Parallel Process Model (EPPM) [20] and Health Belief Model (HBM) [21], which establishes the definitions of fear, threat and barrier based on a profound understanding of their nature [19]. Even though the fear scale was originally developed to measure the fear of breast cancer, a study validated the scale to measure the fear of COVID-19 using BCFS in mainland China and Hong Kong [22]. Moreover, the theoretical framework of BCFS (i.e., EPPM and HBM) has been widely used for research addressing public health emergencies including COVID-19. While several Korean validated COVID-19 fear scales are available [23, 24], the scales were neither validated with a larger Korean population, nor grounded in robust theoretical frameworks.

## 2. Materials and methods

### 2.1. Study design and procedure

This methodology study was guided by the COSMIN Study Design checklist for patient-reported outcome measurement instruments [25]. The psychometric evaluation process consisted of two phases. The first phase focused on the translation of the BCFS, and the face and content validity were assessed. The second phase emphasised the BCFS validation through a psychometric testing process including internal consistency, reliability, convergent validity, known-group validity and item response theory(IRT)-based analysis, using the Korean dataset in the large scale of international survey [26].

## 2.2. Phase I: Translation of the BCFS

We initially translated the BCFS into Korean version. The BCFS that was originally developed to measure psychological response in women with breast cancer and assured its construct validity and reliability (Cronbach's alpha = 0.92) [19] was adopted for this study to measure the fear of COVID-19. The scale consists of 8 items with a 5-point Likert scale response. A higher total score refers to a higher fear level (total score ranging from 8 to 40). The scale also provided cut-off scores for the fear level (low: 8–15; moderate 16–23; and high: 24–40). The term 'breast cancer' in the original BCFS was changed to 'COVID-19' for this study. The forward- and back-translation strategies were performed to translate the scale in English to Korean. Two English-Korean bilingual researchers forward-translated the BCFS(English) to Korean [27]. Then, another bilingual researcher who did not know the original BCSF carried out a back-translation on the Korean version of BCFS to English. Along with the researchers who performed the forward- and back- translations, the research team held a reconciliation meeting to reach a consensus version. Lastly, a native English speaker was also invited to compare the original and back English versions of BCFS.

The content validity of the translated fear scale was evaluated by the research expert panel (including psychological and psychometric research experts; n = 6) of this study. The content relevance, clarity, and the accuracy of translation of the scale were considered to calculate the content validity index (CVI), of which a value ≥80% was used as the assessment standard, the proportion in which ≥80% of the experts agreed with scores ≥3 points (5-point rating while 5 referred to high relevance) on the scale. The suggestions of the experts were considered in modifying the scale, if any. Six experts evaluated the content of the K-FS-8 and the overall CVI calculated for K-FS-8 was 80%. The panel confirmed that no further modification was needed for the K-FS-8. Additionally, cognitive debriefing interviews were conducted with 7 general Korean adults who did not involve in this study as a participant for face validity. The scale was then administered to 7 adults with different sociodemographic characteristics whose education level were primary school in order to assess the face validity, clarity and readability of the translated items [28]. All of the 7 adults stated that the scale was understandable, and no further change was required. Then, the final version of K-FS-8 was pretested in monolingual (Korean) populations. The Korean version of fear scale(K-FS-8) was confirmed and available in the S1 File.

## 2.3. Phase II: Validation of the Korean version of COVID-19 fear scale (K-FS-8)

**2.3.1. Sample for the psychometric testing.** A population-based cross-sectional online survey was conducted in South Korea from 23 August to 13 September 2020 using the largest online survey platform in South Korea (https://embrain.com;>1.5 million Korean panel members nationwide). The platform has annually conducted approximately 4,800 online surveys for academic research, government and industry. The eligibility criteria for participation were 1)adult (≥20 years old according to the civil law in South Korea) and 2)a South Korea resident who can read and understand the Korean language. The survey company randomly sent the study invitations via registered emails to their panel members who met the criteria for 3 weeks (approximately 3,500 emails per week; a total of 10,757 emails were sent). The invitation email included general information regarding the survey. Participants spent approximately 15 min to complete the survey (n.b., participants were required to answer all survey questions to complete the survey, hence no missing data). Meanwhile, the first COVID-19 case was reported on 20th January 2020 in South Korea. Since then, the second wave of COVID-19 occurred from 13 August to 18 September 2020 (this online survey started in the early

part of the second wave). The average daily COVID-19 cases during the period were approximately 220 [15]. During the period, COVID-19 vaccine was unavailable, and strict social distancing measures were conducted in South Korea. Participants' socio-demographics including age, gender, marital status, education level and the number of people living together were also collected.

**2.3.2. Convergent validity, known-group validity, internal consistency and reliability of the K-FS-8.**   The Patient Health Questionnaire-4(PHQ-4; 4 items) [29] and Primary Care Post-Traumatic Stress Disorder Screen for DSM-5 (PC-PTSD-5; 5 items) [30] were adopted to evaluate the convergent validity of the Korean fear scale. Convergent validity indicates to how closely a test is related to other tests that measure similar constructs. The PHQ-4 scale consists of two sub-scales, namely anxiety(2 items) and depression(2 items) [29]. The total score of each sub-scale in PHQ-4 ranges from 0 to 6 and $\geq$3 total scores of each sub-scale refer to having anxiety and depressive symptoms, respectively. The Cronbach's alpha of PHQ-4 was 0.82 (the sub-scales' coefficients = 0.75[anxiety] and 0.78[depression]) [29]. The PC-PTSD-5 scale was developed to screen the posttraumatic stress disorder (PTSD) [30]. The total score ranges from 0 to 5. A higher score of PC-PTSD-5 indicates a higher risk of PTSD, and the cut-off score of PTSD high risk is 3. The test-retest reliability of PC-PTSD-5($r$) was 0.83 [30]. Both PHQ-4 and PC-PTSD-5 scales were previously validated in Korean [31, 32]. The Pearson's correlation coefficient (i)between the total score of the K-FS-8 and total score of the PHQ-4 and (ii)between the total score of the K-FS-8 and total score of the PHQ-4 and PC-PTSD were computed for the convergent validity of the K-FS-8.

For internal consistency, the corrected item-total correlation was adopted to assess the internal construct validity of the scale. A correlation coefficient $\geq$0.4 indicated adequate internal construct validity [33]. To assess the known-group validity, independent $t$-tests were used to compare the mean scores of the K-FS-8 between males and females. To compare the mean scores across the different age groups and marital status, an one-way ANOVA test was adopted. In addition, Cohen's $d$ effect sizes–trivial ($<$0.2), small ($\geq$0.2 and $<$0.5), moderate ($\geq$0.5 and $<$0.8) and large ($\geq$0.8) were computed [34]. Last but not least, the fear levels of the participants were investigated using descriptive statistics.

**2.3.3. Item response theory (IRT)-based analysis.**   Prior to conducting the parametric IRT analysis, confirming the assumptions of unidimensionality is required. Unidimensionality refers to items on an instrument measure in only one concept. For the assessment of unidimensionality, the exploratory factor analysis(EFA) was performed [35, 36]. To determine dimensionality, we used the scree plot and Kaiser's eigenvalue-greater-than-one criteria from the EFA. Parallel analysis was performed to confirm the number of factor extraction [37]. The standardized root mean square residual(SRMSR) and comparative fit index(CFI) were assessed to further confirm the adequacy of model fit. The degree of item fit was assessed using a recommended index, S- $\chi2$, to compute an RMSEA value. A value of less than .06 is deemed an appropriate fit [38, 39]. After the appropriateness of the model and item fit was determined, we estimated item parameters using the generalised partial credit model in this IRT-based analysis. The discrimination and location parameters are generated by the IRT parameterization to assess item-latent trait relationships. The IRT plots including the category characteristic curves, item information curves, scale information and conditional standard error curves, conditional reliability curve and scale characteristic curve were generated to visually investigate the item and the characteristics of scale using various plots. These IRT plots present the relationship among each item, the total scale and the latent trait across trait values [40, 41]. R Statistical Software(version 4.2.1) was used for statistical analyses in this study.

## 2.4. Ethical considerations

Ethical approval for this study was obtained by a university institutional review board(IRB; UW 20–272). Study purpose and participant's rights were provided in the first page of online survey. By clicking the 'agree to participate in this survey' menu(i.e., informed consent), they could start answering the survey questions.

# 3. Results

## 3.1. Participants' characteristics

A total of 2,235 participants(response rate: 20.78%) aged 20–69 years old (mean age: 44.00 years, SD = 13.64) participated in the study. Among them, 49.3% were male and 50.4% were female. In terms of education, 65.5% of the study participants had a Bachelor's degree or above. More than half of the study participants were currently married(59.5%), and more than one-third of the study participants were currently single(35.5%). **Table 1** demonstrates the study participants' characteristics.

## 3.2. Reliability and validity of the K-FS-8

The mean score of the K-FS-8 was 28.57(SD: 5.79), and the Cronbach's alpha was 0.92. The corrected item-total correlations were >0.4 for the items. The Pearson's correlation coefficient between the K-FS-8 and the PHQ-4 scores was 0.47($p<0.01$). It was 0.50 between the score of the K-FS-8 and the score of the PC-PTSD ($p<0.01$). **Table 2** demonstrates the internal consistency, internal construct validity and convergent validity results. The results of the independent $t$-tests in the known-group comparisons indicated that female had a higher fear level than

**Table 1. Socio-demographic profile.** (N = 2,235).

| | |
|---|---|
| Mean age of study participants (SD) | 44.00 years (13.64) |
| *20–29 years* | 450 (20.1%) |
| *30–39 years* | 447 (20.0%) |
| *40–49 years* | 451 (20.2%) |
| *50–59 years* | 444 (19.9%) |
| *60–69 years* | 443 (19.8%) |
| Gender | |
| *Male* | 1102 (49.3%) |
| *Female* | 1126 (50.4%) |
| *Non-binary* | 7 (0.3%) |
| Marital status | |
| *Single* | 793 (35.5%) |
| *Married/Cohabitation* | 1330 (59.5%) |
| *Separated/ divorce/ widowed* | 112 (5.0%) |
| Education | |
| *Below Bachelor's degree* | 768 (34.4%) |
| *Bachelor's degree or the above* | 1467 (65.6%) |
| Number of people living together | |
| *Alone* | 268 (12.0%) |
| *1* | 428 (19.2%) |
| *2* | 638 (28.5%) |
| *3* | 755 (33.8%) |
| *≥4* | 146 (6.5%) |

**Table 2. Descriptive statistics, convergent validity and reliability.**

| | Corrected Item-Total Correlation | Mean SD | Strongly Disagree | Disagree | Neutral | Agree | Strongly Agree | Strongly Disagree/ Disagree | Strongly Agree/ Agree |
|---|---|---|---|---|---|---|---|---|---|
| The thought of COVID-19 scares me | 0.72 | 3.85 (0.83) | 28 (1.3%) | 124 (5.5%) | 421 (18.8%) | 1249 (55.9%) | 413 (18.5%) | 152 (6.8%) | 1662 (74.4%) |
| When I think about COVID-19, I feel nervous | 0.75 | 3.74 (0.84) | 34 (1.5%) | 153 (6.8%) | 507 (22.7%) | 1217 (54.5%) | 324 (14.5%) | 187 (8.4%) | 1541 (68.9%) |
| When I think about COVID-19, I get upset | 0.64 | 3.91 (0.87) | 32 (1.4%) | 111 (5.0%) | 424 (19.0%) | 1129 (50.5%) | 539 (24.1%) | 143 (6.4%) | 1668 (74.6%) |
| When I think about COVID-19, I get depressed | 0.77 | 3.52 (0.98) | 72 (3.2%) | 250 (11.2%) | 686 (30.7%) | 899 (40.2%) | 328 (14.7%) | 322 (14.4%) | 1227 (54.9%) |
| When I think about COVID-19, I get jittery | 0.71 | 3.27 (0.98) | 97 (4.3%) | 339 (15.2%) | 875 (39.1%) | 711 (31.8%) | 213 (9.5%) | 436 (19.5%) | 924 (41.3%) |
| When I think about COVID-19, my heart beats faster | 0.66 | 2.85 (0.97) | 217 (9.7%) | 485 (21.7%) | 1047 (46.8%) | 378 (16.9%) | 108 (4.8%) | 702 (31.4%) | 486 (21.7%) |
| When I think about COVID-19, I feel uneasy | 0.69 | 3.82 (0.86) | 43 (1.9%) | 127 (5.7%) | 410 (18.3%) | 1255 (56.2%) | 400 (17.9%) | 170 (7.6%) | 1655 (74.0%) |
| When I think about COVID-19, I feel anxious | 0.81 | 3.61 (0.93) | 59 (2.6%) | 203 (9.1%) | 602 (26.9%) | 1061 (47.5%) | 310 (13.9%) | 262 (11.7%) | 1371 (61.3%) |
| Cronbach's Alpha | 0.92 | | | | | | | | |
| PTSD total score | | 1.62 (1.49) | High risk of PTSD = 588 (26.3%) | | | | | | |
| PHQ-4 total score | | 2.92 (2.81) | | | | | | | |
| • Anxiety symptom total score | | 1.23 (1.44) | Anxiety symptom = 300 (13.4%) | | | | | | |
| • Depressive symptom total score | | 1.69 (1.62) | Depressive symptom = 518 (23.2%) | | | | | | |
| K-FS-8 total score | | 28.57 (5.79) | High fear = 1891 (84.6%); Moderate fear = 290 (13.0%); Low fear = 54 (2.4%). | | | | | | |
| | | PTSD total score | PHQ-4 total score | | | | | | |
| Correlation coefficient | K-FS-8 | 0.498** | 0.467** | | | | | | |

male(Cohen's $d$: 0.48). A one-way ANOVA test showed that the fear score differed across the age and marital status. Study participants in the age groups 30–39, 40–49, 50–59 and 60–69 years had a higher fear level than the participants in the age group of 20–29 years(Cohen's $d$: 0.30, 0.38, 0.30 and 0.23, respectively). In addition, the study participants who were married had a higher fear level than the participants who were single(Cohen's $d$: 0.24). **Table 3** demonstrates the results of the known-group comparisons.

### 3.3. Levels of fear, PTSD and anxiety and depressive symptoms in the participants

The results showed that the participants had high and moderate levels of COVID-19 fear(84.6% and 13.0%, respectively). **Table 2** shows the descriptive statistics of each K-FS-8 item. In addition, 13% and 23.2% of participants reported anxiety and depressive symptoms, respectively, whilst 26.3% had a high risk of PTSD during the 2nd wave of COVID-19 pandemic in South Korea.

### 3.4. Item response theory (IRT)-based analysis

**3.4.1. Model assumption: EFA results.** One factor was extracted via EFA with eigenvalues of >1(5.111; Table 4) for the assumption of unidimensionality, accounting for a total of

**Table 3. Known-group comparison.**

| | Male | Female | | | | | | | | |
|---|---|---|---|---|---|---|---|---|---|---|
| | **n = 1102** | **n = 1126** | | | | | | | | |
| | **Mean (SD)** | **Mean (SD)** | **p-value** | **Cohen's D effect Size** | | | | | | |
| K-FS-8 | 27.19 (5.85) | 29.92 (5.40) | <0.01 | 0.48 | | | | | | |
| | **20–29** | **30–39** | **40–49** | **50–59** | **60–69** | | | | | |
| | **n = 450** | **n = 447** | **n = 451** | **n = 444** | **n = 443** | | | | | |
| | *1* | *2* | *3* | *4* | *5* | | | | | |
| | **Mean (SD)** | **Mean (SD)** | **Mean (SD)** | **Mean (SD)** | **Mean (SD)** | **p-value ^** | ***Cohen's D Effect size (1 vs. 2)*** | ***Cohen's D Effect size (1 vs. 3)*** | ***Cohen's D Effect size (1 vs. 4)*** | ***Cohen's D Effect size (1 vs. 5)*** |
| K-FS-8 | 27.18 (5.87) | 28.89 (5.71) | 29.27 (5.11) | 28.94 (5.79) | 28.56 (6.23) | <0.01 | 0.30 | 0.38 | 0.30 | 0.23 |
| | **Single** | **Married** | **Separated** | | | | | | | |
| | **n = 793** | **n = 1330** | **n = 112** | | | | | | | |
| | *1* | *2* | *3* | | | | | | | |
| | **Mean (SD)** | **Mean (SD)** | **Mean (SD)** | **p-value ^** | ***Cohen's D Effect size (1 vs. 2)*** | | | | | |
| K-FS-8 | 27.67 (5.97) | 29.08 (5.61) | 28.84 (5.88) | <0.01 | 0.24 | | | | | |

^ one-way ANOVA with post-hoc test

63.882% variance. The factor with item loading ranges from 0.722 to 0.867. The parallel analysis with the scree plot suggested a single factor. Therefore, the unidimensionality was confirmed (Fig 1).

**3.4.2. Model fit.** The SRMSR value = 0.072 indicates that the data fit the model satisfactorily using the suggested cut-off values of SRMSR≤0.08 as criteria for assessing fit. The CFI = 0.928 barely fell short of the recommended 0.95 threshold.

**3.4.3. IRT item fit and IRT parameters.** All RMSEA values were below 0.06, suggesting that the items fit the model well. Table 5 displays the estimated IRT parameters. The slope parameters(a-parameters) had values ranging from 1.78 to 3.69. A slope parameter quantifies how well an item distinguishes respondents with varying degrees of the latent trait. Steeper

**Table 4. EFA model and factor loading.**

| Item | Factor loading | Communalities |
|---|---|---|
| q1 The thought of COVID-19 scares me | 0.798 | 0.637 |
| q2 When I think about COVID-19, I feel nervous | 0.823 | 0.677 |
| q3 When I think about COVID-19, I get upset | 0.722 | 0.521 |
| q4 When I think about COVID-19, I get depressed | 0.829 | 0.687 |
| q5 When I think about COVID-19, I get jittery | 0.844 | 0.712 |
| q6 When I think about COVID-19, my heart beats faster | 0.739 | 0.546 |
| q7 When I think about COVID-19, I feel uneasy | 0.761 | 0.579 |
| q8 When I think about COVID-19, I feel anxious | 0.867 | 0.751 |
| Eigenvalue = 5.111 | | |
| Explanation (accumulated) variation = 63.882% | | |

*Note.* The obtained KMO value of 0.900 (> 0.5) and results of Barlett's test of sphericity (p < .0001) suggested that the collected data were appropriate for EFA.

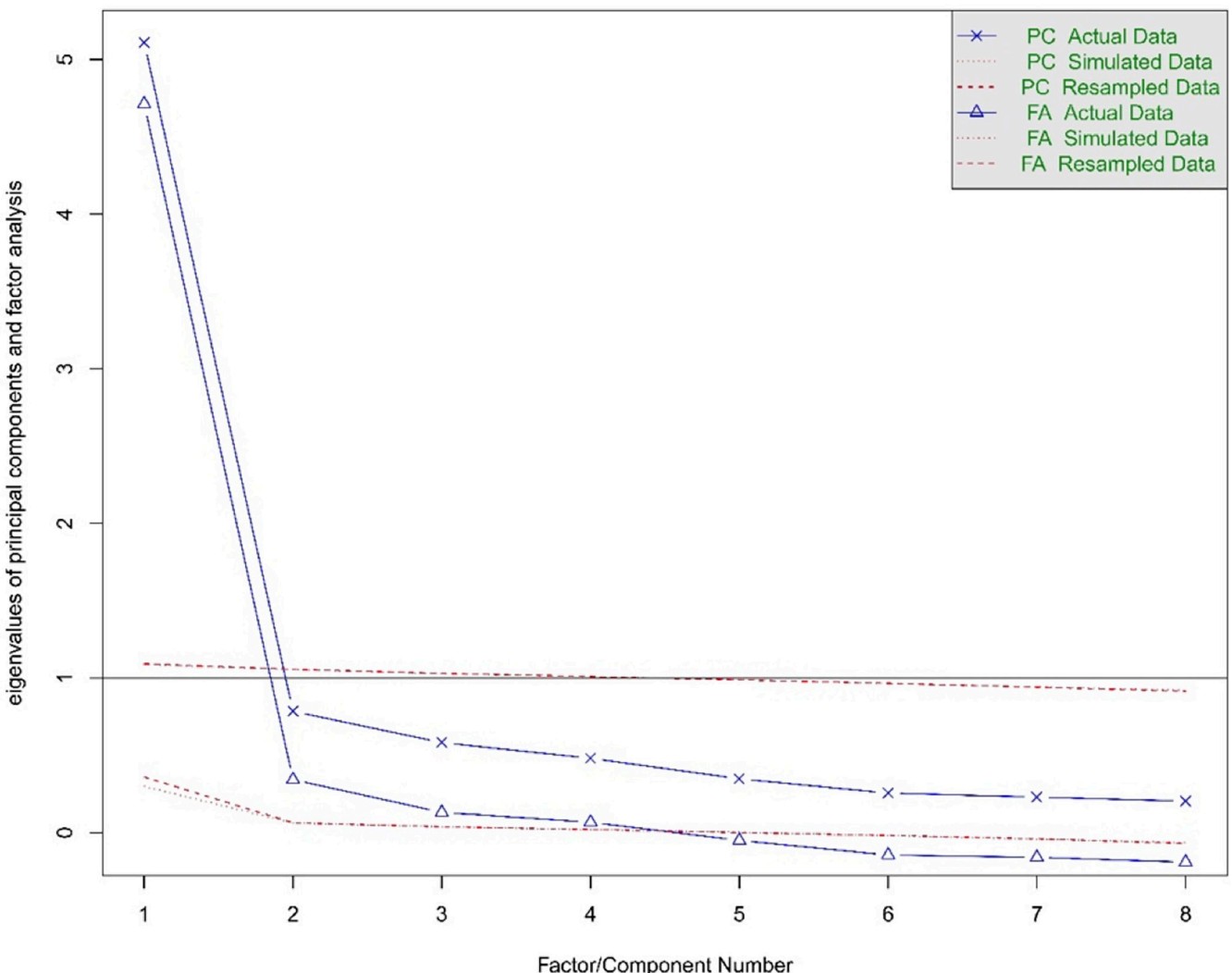

**Fig 1. Parallel analysis of the K-FS-8.**

slopes or larger values are more effective in distinguishing theta. A slope can also be used to determine the strength of a link between an item and a latent characteristic, with greater slope values indicating stronger associations. Item q8 had the highest slope estimate of 3.69, whereas Item 3 had the lowest with a slope estimate of 1.78. For each item, location parameters(b-parameters) are also mentioned.

**3.4.4. Category characteristic curves.** Each symmetrical curve shows the likelihood of supporting a response category(P1, P2, P3, P4, and P5 refer to 'strongly disagree', 'disagree', 'neutral', 'agree', and 'Strongly agree' respectively). The curves show a functional relationship with theta–when theta increases, the likelihood of supporting a category also increases, which subsequently decreases when answers shift to the next higher category. According to the categorical response curves(CRCs), the response categories encompass a wide range of theta (Fig 2).

**3.4.5. Item information curves.** Fig 3 shows item information curves of the K-FS-8. Information is a statistical concept that pertains to an item's capacity to predict theta scores

**Table 5. Item characteristics, item fit and IRT parameters of the K-FS-8.**

| | Item fit | | | | IRT parameters | | | | |
|---|---|---|---|---|---|---|---|---|---|
| Item | S-$\chi^2$ | df | RMSEA.S-$\chi^2$ | p. S-$\chi^2$ | a | b1 | b2 | b3 | b4 |
| q1 | 140.98 | 36 | 0.03 | 0.000 | 2.71 | -1.89 | -0.82 | 1.00 | NA |
| q2 | 79.37 | 32 | 0.02 | 0.000 | 2.98 | -3.04 | -1.69 | -0.62 | 1.17 |
| q3 | 238.33 | 45 | 0.04 | 0.000 | 1.78 | -2.32 | -0.94 | 0.92 | NA |
| q4 | 88.39 | 40 | 0.02 | 0.000 | 2.70 | -2.60 | -1.36 | -0.21 | 1.20 |
| q5 | 119.04 | 42 | 0.02 | 0.000 | 2.83 | -2.32 | -1.10 | 0.15 | 1.52 |
| q6 | 242.09 | 50 | 0.04 | 0.000 | 1.80 | -2.05 | -0.78 | 0.96 | 2.36 |
| q7 | 142.74 | 41 | 0.03 | 0.000 | 2.15 | -1.99 | -0.86 | 1.12 | NA |
| q8 | 56.27 | 32 | 0.01 | 0.005 | 3.69 | -2.55 | -1.40 | -0.38 | 1.13 |

*Note.* α indicates discrimination parameters. *b1–b4* reflects location parameters. Df = degree of freedom. S −$\chi^2$ statistic was used to examine the test of item fit and all items exhibited acceptable fit as alpha values were > 0.001.

properly. Item level information emphasises how effectively each item contributes to the precision of score estimation, with increasing levels of information resulting in more accurate score predictions. The quantity of information an item contributes to polytomous models is determined by its slope parameter—the bigger the value, the more information the item delivers. Furthermore, the more information the item provides, the further apart the location parameters of b1, b2, b3, and b4. Over theta, an ideally informative polytomous item would often have a broad category coverage and large location, as indicated by location parameters.

The additional IRT plots, including curves of 1) scale information and conditional standard errors, 2) conditional reliability curve and 3) scale characteristic curve can be found in the S2 File.

## 4. Discussion

Using a large sample of Korean adults, the study evaluated the psychometric properties of the K-FS-8 to ensure that the scale is valid (convergent validity, known-group validity and IRT analysis) and reliable(Cronbach's α coefficient = 0.92; high coefficient) in measuring the severity of COVID-19 fear among the Korean general population. The results support the general agreement between the 8 items that comprise the composite score of the scale to measure the fear related to COVID-19. All the item-total correlation coefficients corrected for overlaps were larger than 0.6. Therefore, the internal construct validity of the modified scale can be supported. Accordingly, the suggestion that all individual items evenly measured the construct was supported by the result. In terms of the study's convergent validity, the correlations between the total score of the K-FS-8 and the total scores of PTSD and PHQ-4 were moderate as 0.498 and 0.467, respectively.

The BCFS was previously validated in Chinese for COVID-19 fear measurement [22]. Similar to this study, the validation study of Choi and colleagues confirmed their convergent validity of BCFS for COVID-19 fear with the PHQ-4; correlations between the level of fear and anxiety and depressive symptoms were validated [22]. This study additionally validated the fear scale with the PTSD score. Considering the importance of associations among psychological distresses such as fear, anxiety and PTSD, the results of this study could be considered an expansive contribution to the validation of BCFS for COVID-19 fear. A more robust and thorough psychometric evaluation was also further undertaken for the Korean version of fear scale (i.e., K-FS-8) through the IRT-based analysis.

The internal consistency(Cronbach's α coefficient) of the K-FS-8 was similar to the Chinese version of the scale(Korean version: 0.92 and Chinese version: 0.93; indicating excellent

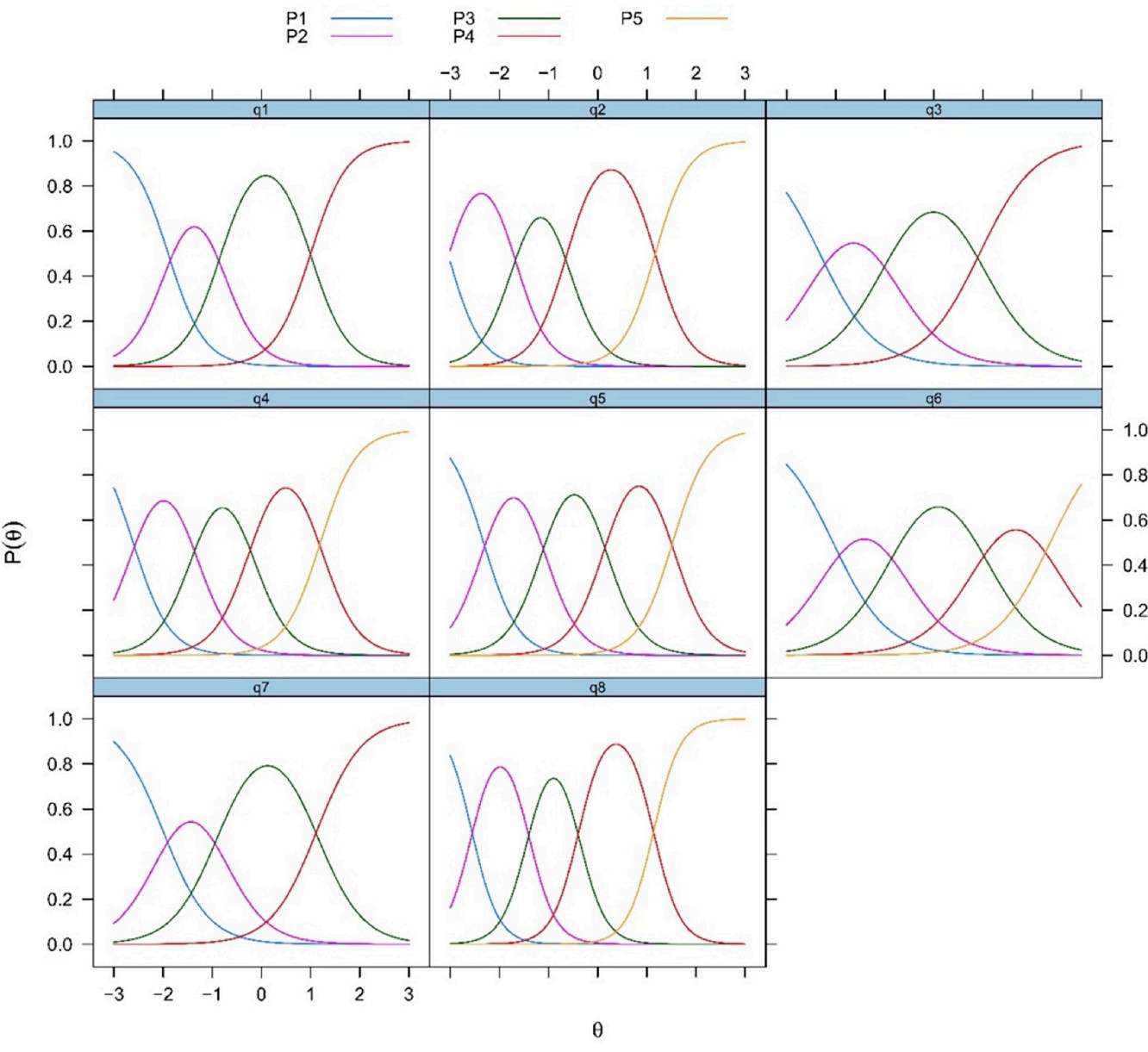

**Fig 2. Category characteristic curves of the K-FS-8.**

reliability) [22]. Therefore, it can be confirmed that BCSF, which was modified for measuring COVID-19 fear both in Korean and Chinese, is valid and reliable. Apart from the BCFS for COVID-19 fear, Ahorsu, et al. [42] also developed the Fear of COVID-19 scale(FCV-19; 7 items), which has been validated in several countries including South Korea. Those studies validated the FCV-19 by measuring the Cronbach's α coefficient which ranged from 0.80 to 0.88 [23, 43–47]. Even though the Cronbach's α coefficient of the validation of FCV-19 shows high consistency, this study validated that BCFS on the fear of COVID-19 has the advantage of higher consistency of 0.92 Cronbach's α coefficient. Another advantage of this validation study is the even distribution of participant age groups. In the case of FCV-19, validation studies targeted the relatively young generation in their 20s to 50s given that the initial scale [42] targeted

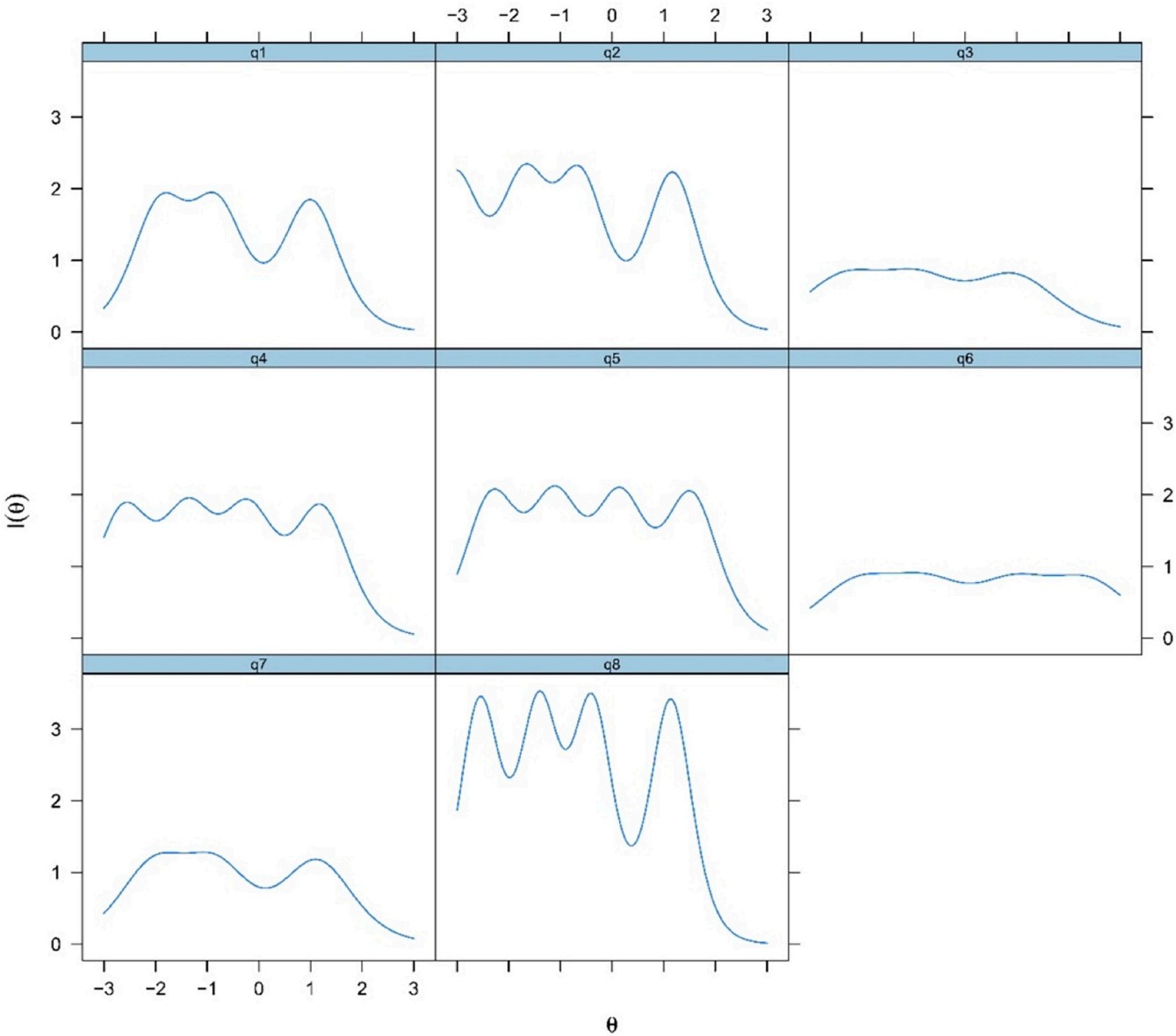

**Fig 3. Item information curves of the K-FS-8.**

the young generation. Particularly, a study conducted in Japan validated FCV-19 in adolescent participants [45]. Therefore, this study that included wider age groups of participants would have advantage. Moreover, this study and the Chinese validation study [22] conducted the psychometric evaluation of BCSF for COVID-19 fear with large samples(n = 2,235 and n = 2,822, respectively), comparing with the FCV-19 studies, which can be another advantage of this study.

Using K-FS-8, this study identified that 84.6% of participants had the fear of COVID-19 during the 2nd wave of COVID-19(mean±SD = 28.57±5.79). The mean score of K-FS-8 is higher than Mainland China(22.40±6.49) and Hong Kong(23.98±6.64) [22]. As previously mentioned, while fear can aid compliance to public health measures in a pandemic, higher levels of fear can precipitate increased stress levels and anxiety, triggering or exacerbating adverse

health behaviours such as smoking [48] and excessive alcohol use [49]. Fear of COVID-19 can not only trigger domestic violence as a stress response, but can prevent victims from seeking help or medical attention for fear of contracting COVID-19 at hospitals [50]. Hence, there is greater significance in screening for people with higher fear levels due to the greater risk of them perpetuating adverse social and health behaviours, warranting psychological intervention as a means of primary and secondary prevention. Examples of effective psychological intervention include techniques in improving appraisal of the body, emotion regulation and attachment security, and adopting acceptance [51]. In this study, prevalence of COVID-19 fear in South Korea (84.6%) is almost twice to four times more than the prevalence of fear of COVID-19 in over 13 other countries (18.1% to 45.2%) in a scoping review study [52]. This indicates that even after stratification into levels of fear-related risk, there would be a greater number of South Korean people with clinically-significant fear requiring intervention, validating the need for proactive screening to identify them.

This study has several limitations. The fear scale(i.e., BCFS) was initially designed for the fear of breast cancer [19]. However, BCFS was developed on the basis of a theoretical framework—the combination of EPPM and HBM—which have been widely used for healthcare practice and research and hence can be applied for other diseases and/or populations. As this study only recruited adults, further validation in younger population would be helpful to achieve generalisable results. The low response rate in this study (i.e., 20.78%) might not represent the target survey population, which would cause potential nonresponse bias. The participants' higher tertiary education attainment (69.3%) was slightly higher than the Korean general population (51.7% in an age group of 25 and 64-year-olds) [53]. The skewed distribution of the education level may potentially influence generalisability of the results. Although K-FS-8 showed a higher internal consistency of Cronbach's α coefficient than existing COVID-19 fear scales, the finding should be interpreted with caution as additional reliability tests such as test-retest reliability was not performed. Lastly, the online survey method of data collection had a potential bias in the sample due to exclusion of the population without computer literacy.

## 5. Conclusion

The 8-item user-friendly Korean version of fear scale(K-FS-8) was validated through a representative sample in South Korea in this study. The scale can be used in measuring fear levels. Additionally, this study identified a high level of COVID-19 fear in the study population. Therefore, healthcare professionals are required to proactively screen their client and the public and subsequently provide psychological interventions for fear due to its adverse health effects. Healthcare professionals can use the K-FS-8 to screen fear of the COVID-19 and related major public health crises with effectiveness and efficiency among their clients and the general public in community and primary care settings to achieve early recognition of clinical deterioration in psychological well-being and health behaviours.

## Supporting information

**S1 File. The Korean version of fear scale (K-FS-8).**
(DOCX)

**S2 File. The additional IRT plots.**
(DOCX)

**S3 File. STROBE checklist.**
(DOCX)

**S1 Fig. Scale information and conditional standard errors.**
(TIFF)

**S2 Fig. Scale characteristic curve.**
(TIFF)

## Author Contributions

**Conceptualization:** Jung Jae Lee, Daniel Y. T. Fong, Kris Yuet Wan Lok, Mandy Ho, Chia-Chin Lin.

**Data curation:** Jung Jae Lee, Hye Ri Choi, Mu-Hsing Ho.

**Formal analysis:** Jung Jae Lee, Hye Ri Choi, Edmond Pui-Hang Choi, Mu-Hsing Ho.

**Investigation:** Jung Jae Lee.

**Methodology:** Jung Jae Lee, Hye Ri Choi, Edmond Pui-Hang Choi, Mu-Hsing Ho, Kris Yuet Wan Lok, Mandy Ho, Chia-Chin Lin.

**Project administration:** Jung Jae Lee.

**Supervision:** Daniel Y. T. Fong, Chia-Chin Lin.

**Validation:** Edmond Pui-Hang Choi, Mu-Hsing Ho, Daniel Y. T. Fong, Mandy Ho.

**Writing – original draft:** Jung Jae Lee, Hye Ri Choi, Mu-Hsing Ho.

**Writing – review & editing:** Jung Jae Lee, Hye Ri Choi, Edmond Pui-Hang Choi, Mu-Hsing Ho, Daniel Y. T. Fong, Kris Yuet Wan Lok, Chia-Chin Lin.

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
