## [Decision Letter · Decision Letter 0]

8 Jan 2023

PONE-D-22-30798Psychometric Evaluation of Korean version of COVID-19 Fear Scale (K-FS-8): a population based cross-sectional studyPLOS ONE

Dear Dr. Fong Daniel Yee-Tak

Thank you for submitting your manuscript to PLOS ONE. After careful consideration, we feel that it has merit but does not fully meet PLOS ONE’s publication criteria as it currently stands. Therefore, we invite you to submit a revised version of the manuscript that addresses the points raised during the review process.

The manuscript has can add significant knowledge in scientific community as it has covered the large group of people. However, authors need to justify the real need of validation of Breast Cancer Fear Scale as there is already available of COVID-19 fear scale in Korean Language. Another minor correction for this statement “Two English-Korean bilingual researchers forward-translated the BCFS(English) to Korean [27]” needs to be checked and restate as the citation provided is confusing. Thus, considering to explain the need of validating this scale to measure fear of COVID-19, you are encouraged to address the reviewers’ comments to make the manuscript publishable 

We look forward to receiving your revised manuscript.

Kind regards,

Bimala Panthee

Academic Editor

PLOS ONE

Journal Requirements:

None. 

5. Please ensure that you refer to Figure 2 in your text as, if accepted, production will need this reference to link the reader to the figure.

Reviewers' comments:

Reviewer's Responses to Questions

**Comments to the Author**

1. Is the manuscript technically sound, and do the data support the conclusions?

Reviewer #1: Partly

Reviewer #2: Yes

Reviewer #3: Yes

Reviewer #4: Partly

2. Has the statistical analysis been performed appropriately and rigorously? 

Reviewer #1: Yes

Reviewer #2: Yes

Reviewer #3: Yes

Reviewer #4: I Don't Know

3. Have the authors made all data underlying the findings in their manuscript fully available?

Reviewer #1: No

Reviewer #2: No

Reviewer #3: No

Reviewer #4: Yes

4. Is the manuscript presented in an intelligible fashion and written in standard English?

Reviewer #1: Yes

Reviewer #2: Yes

Reviewer #3: Yes

Reviewer #4: Yes

5. Review Comments to the Author

Reviewer #1: December 19, 2022

Thank you for the opportunity to review the paper “Psychometric evaluation of Korean version of COVID-19 fear scale (K-FS-8): a population based cross sectional study”, which I thoroughly enjoyed. The authors attempt to translate and validate the COVID-19 fear scale in Korean population by adapting a fear scale designed originally for breast cancer focusing on multiple aspects of validity.

By validating the scale, the authors make a valuable addition to scales existent in mental health research, in context of Korea. However, I believe the manuscript would benefit by considering the following points. Please consider the comments as constructive and also note that I do not have expertise in psychometrics.

Abstract

Please consider not using acronyms in the abstract. Save it later for the manuscript where the terms are first used.

Introduction

I believe that the introduction would benefit from adding on to the literature on existence of other scales for measuring fear during covid- 19, commonly in use globally and in context of Korea. A quick search yielded many interesting publications with covid- 19 fear scale, the most widely used and it appears that this has been validated in Korea as well in 2021.

Please refer to the following few articles for the reference:

The Korean Version of Fear of COVID-19 Scale: Psychometric Validation in the Korean Population - PubMed (nih.gov)

Instruments to measure fear of COVID-19: a diagnostic systematic review | BMC Medical Research Methodology | Full Text (biomedcentral.com)

Further, the introduction can be shortened in their first and third paragraphs.

The authors mention no validated fear scales for COVID-19 exists in Korea as the major rationale for the conduct of the study, though they have included this in the discussion section. Please justify the statement on what the authors meant.

Also, I suggest the authors be explicit in why they wanted to validate the scale that was originally intended for breast cancer. Is there any theory to guide?

Materials and Methods

The authors mention that face validity and cognitive interviewing were done in phase 1. Do the authors have any data to demonstrate the content validity and what changes were made in the original translation after these steps? How were the findings analyzed? It would be helpful if the authors provide a table to the demographics on whom cognitive interviewing was done for validity of the findings.

I understand this was a survey, but based on research question, did the authors have any estimation for sample size prior to conducting the study?

The authors state that PHQ-4 and PC-PTSD-5 were adopted for this study to assess convergent validity. What do the authors mean with this statement? Were these instruments not already validated in the Korean population?

For known- group validity, T test and ANOVA have been used. I wonder if the authors checked for the distribution of the data before going with the tests. Please clarify.

Item response theory (IRT)- were the assumptions for IRT met?

Results

Were there any missing data on any items? Please clarify.

Discussion

The discussion would benefit by including more literature in relation to fear scales in the local and the global context and the limitations with the existing scales.

The authors mention that fear might be higher in Korean sample. What do the authors mean?

Also, the fact that fear is a natural response to any threat, perceived or real and is protective. So, high level of fear during covid-19 at a time when nothing was known about it is justified. I wonder why the authors focus on the statement that high level of fear means it always needs intervention. As also reported in the conclusion. I believe the discussion should focus more on clinical or morbid fear that necessitates intervention, by doing a more through investigation. I suggest the authors consider this in discussion as well as conclusion.

The authors have pointed out the limitations with the study, which is good.

A critical aspect is that even if the correlations are high and indicate reliability, they might not be a valid indicator of what we intend to measure. So, this is also a limitation to studies which focus on one aspect of validity. The authors can consider this point also to highlight in their discussion. Other aspects such as divergent validity could also be discussed.

I wish the authors all the best.

Reviewer #2: The manuscript is excellent in terms of the factor analysis and model fit calculations. However, there are concerns about the theoretical foundations that require to be ironed out. The study attempts to validate a fear scale based on a non-communicable disease (breast cancer) for use in a context of a communicable disease (COVID-19), whiile the statistical analysis is sound, it is recommended that the authors strenghthen their justification by communicating with the original authors of the BFCS to discuss the interchangeable nature of the scale, or provide and describe examples from the literature where this has been previously accomplished. More comments included in the attached document.

Reviewer #3: Dear authors, this paper is interesting. However, it aims to explore the Psychometric Evaluation of the Korean version of the COVID-19 Fear Scale (K-FS-8). Before accepting this paper, significant corrections should be applied.

Abstract:

1- Online survey by which program.

2- PTSD and IRT scales which language that you use?

3- Only translation and convergent validity were used. Please add KMO test and Bartlett test of sphericity. Explanation

4- you measure only Cronbach's alpha for reliability. please calculate compost also.

5- Add total variance to your findings.

editing, please.

Introduction

1- Nice introduction, but you mention the aim of your study at the end. Any related scales (fear) globally were not published before.

2- Write down several validated scales to measure phycological concepts before your aims and then give your readers why this scale is important among the Korean population.

3- Please check your references (Endnote) Champion, CS Skinner, U Menon, S Rawl, RB Giesler, P Monahan, and J Daggy

Methods

1- COSMIN Study Design what is this? More Explanation

2- A 6 expert what are the results, and what about the Important Score?

3- Did they make any modifications to fear items? It is not clear

4- Inclusion criteria. above 20 years old. How you can detect that 15 years cannot participate or the same person can repeat the scale again and again? Also, did you add a consent form?

5- Change sex to gender, please.

Analysis

1- Please add KMO, and Bartlet test.

2- Perfect analysis

Discussion

1- Add what this study will add.

2- Compare what happened before COVID-19 and during it.

3- We do not prefer numbers on discussion.

4- Add implications.

References need critical revisions.

Good luck

Reviewer #4: Comments to the Author

Authors have written a very clear, valuable, and extensive paper with significant number of subjects involved. I congratulate them on this piece of work.

I have some minor comments for the authors to consider.

It was very hard to mark parts of the manuscript because there were neither rows nor page numbers in the submitted manuscript.

I will do my best to be as clear as possible.

In the 2.3.1. Sample for the psychometric testing

You stated that “The eligibility criteria for participation were 1) adult (≥20 years old according to the civil law in South Korea) and 2) a South Korea resident who can read and understand the Korean language.”

How can you be sure that participants in your study were Korean residents and that they were fluent in the Korean language? Please explain.

In the 2.3.2. Convergent validity, known-group validity, internal consistency and reliability of the K-FS-8

You stated that: “The test-retest reliability of PC-PTSD-5(r) was 0.83.”

Was this result calculated in your study? if not, please add the reference.

In the 3.1. Participants’ characteristics

You stated that response rate was 20.78%.

Is this an acceptable percentage for studies like this? Please provide reference to confirm that this percentage is acceptable. In addition, please, discuss how this low percentage of response could influence your results in limitation section.

In the same section as above, you stated that 65.5% of the study participants had a Bachelor’s degree or above. What is the average educational level in Korea? Is sample in this study representative of the Korean population? If not, please discuss how high percentage of highly educated subjects could influence your results.

Please revise the percentage calculation in Table 1 (socio-demographic profile) for education and the number of people living together.

6. PLOS authors have the option to publish the peer review history of their article (what does this mean?). If published, this will include your full peer review and any attached files.

Reviewer #1: **Yes: **Saraswati Dhungana

Reviewer #2: No

Reviewer #3: No

Reviewer #4: No

---

## [Author Response · Author response to Decision Letter 0]

9 Feb 2023

Dear Reviewers,

Thank you very much for offering constructive comments. We have seriously considered the reviewers’ comments and carefully revised the manuscript. Please find our responses to the reviewers’ comments point by point below.

Reviewer #1

R1-1. Thank you for the opportunity to review the paper “Psychometric evaluation of Korean version of COVID-19 fear scale (K-FS-8): a population based cross sectional study”, which I thoroughly enjoyed. The authors attempt to translate and validate the COVID-19 fear scale in Korean population by adapting a fear scale designed originally for breast cancer focusing on multiple aspects of validity.

By validating the scale, the authors make a valuable addition to scales existent in mental health research, in context of Korea. However, I believe the manuscript would benefit by considering the following points. Please consider the comments as constructive and also note that I do not have expertise in psychometrics.

RESPONSE: Thank you very much for your constructive comments. Please find our responses to your comments as follows.

R1-2. Abstract: Please consider not using acronyms in the abstract. Save it later for the manuscript where the terms are first used.

RESPONSE: We have removed all acronyms in the abstract, excepting COVID-19 and K-FS-8. 

R1-3. Introduction: I believe that the introduction would benefit from adding on to the literature on existence of other scales for measuring fear during covid- 19, commonly in use globally and in context of Korea. A quick search yielded many interesting publications with covid- 19 fear scale, the most widely used and it appears that this has been validated in Korea as well in 2021.

Please refer to the following few articles for the reference:

The Korean Version of Fear of COVID-19 Scale: Psychometric Validation in the Korean Population - PubMed (nih.gov)

Instruments to measure fear of COVID-19: a diagnostic systematic review | BMC Medical Research Methodology | Full Text (biomedcentral.com)

Further, the introduction can be shortened in their first and third paragraphs.

The authors mention no validated fear scales for COVID-19 exists in Korea as the major rationale for the conduct of the study, though they have included this in the discussion section. Please justify the statement on what the authors meant.

RESPONSE: Thank you for the constructive comment. In consideration of your comment, we have amended the sentences by adding the existing Korean studies as references and justifying the reason to conduct this study, as follows: 

“However, fear of COVID-19 in South Korea has not been investigated using a large sample. Therefore, this study aimed to measure the COVID-19 related fear with a large sample of Korean population during COVID-19 pandemic, by conducting a psychometric evaluation of the Korean version of fear scale. This study adopted the Breast Cancer Fear Scale (BCFS; 8 items) [19] for the psychometric evaluation. The BCFS was developed on the robust theoretical frameworks – Extended Parallel Process Model (EPPM)[20] and Health Belief Model (HBM)[21], which establishes the definitions of fear, threat and barrier based on a profound understanding of their nature[19]. Even though the fear scale was originally developed to measure the fear of breast cancer, a study validated the scale to measure the fear of COVID-19 using BCFS in mainland China and Hong Kong[22]. Moreover, the theoretical framework of BCFS (i.e., EPPM and HBM) has been widely used for research addressing public health emergencies including COVID-19. While several Korean validated COVID-19 fear scales are available [23, 24], the scales were neither validated with a larger Korean population, nor grounded in robust theoretical frameworks.”

In addition, further comparisons between the fear scale we validated, and the existing validated scales were discussed in Discussion section. 

We have also shortened the first and third paragraphs in Introduction section. 

R1-4. Also, I suggest the authors be explicit in why they wanted to validate the scale that was originally intended for breast cancer. Is there any theory to guide?

RESPONSE: Yes. Compared to the existing COVID-19 fear scales, the original breast cancer fear scale (BCFS) that was validated in this study for measuring COVID-19 fear (K-FS-8) was developed on the robust theoretical framework—Extended Parallel Process Model (EPPM) and Health Belief Model (HBM). Moreover, although the BCFS was initially developed to measure the fear of breast cancer, a study validated for measuring the fear of COVID-19 using BCFS in mainland China and Hong Kong. We have added the description of theoretical underpinning as follows: 

“This study adopted the Breast Cancer Fear Scale (BCFS; 8 items) [19] for the psychometric evaluation. The BCFS was developed on the robust theoretical frameworks – Extended Parallel Process Model (EPPM)[20] and Health Belief Model (HBM)[21], which establishes the definitions of fear, threat and barrier based on a profound understanding of their nature[19]. Even though the fear scale was originally developed to measure the fear of breast cancer, a study validated the scale to measure the fear of COVID-19 using BCFS in mainland China and Hong Kong[22].”

Additionally, we have also discussed the use of BCFS for COVID-19 fear in Discussion section as follows: 

“The fear scale(i.e., BCFS) was initially designed for the fear of breast cancer[21]. However, BCFS was developed on the basis of a theoretical framework—the combination of EPPM and HBM—which have been widely used for healthcare practice and research and hence can be applied for other diseases and/or populations.”

R1-5. Materials and Methods: The authors mention that face validity and cognitive interviewing were done in phase 1. Do the authors have any data to demonstrate the content validity and what changes were made in the original translation after these steps? How were the findings analyzed? It would be helpful if the authors provide a table to the demographics on whom cognitive interviewing was done for validity of the findings.

RESPONSE: Thank you, we have clarified the face validity (assessed by 7 Korean adults) and content validity (evaluated by research expert panel, n=6) mentioned in the phase 1. The content validity index (CVI) was generated using a five-point Likert-type assessment table (5 = highly relevant/clear/accurate) according to the evaluation of the expert panel on the content relevance, clarity, and the accuracy of translation for each item on the scale (Haynes et al 1995). In this stage, no wording or translation issue was identified with a CVI = 0.8, which met the assessment standard. Then, face validity was assessed by 7 Korean adults with a primary school education level to confirm the clarity and readability of the translated items. Again, no further change was required, and the final version of the translated scale was confirmed and prepared for the further psychometric evaluation. We have added further explanation in the text as follows: 

“The content validity of the translated fear scale was evaluated by the research expert panel (including psychological and psychometric research experts; n=6) of this study. The content relevance, clarity, and the accuracy of translation of the scale were considered to calculate the content validity index (CVI), of which a value ≥80% was used as the assessment standard, the proportion in which ≥80% of the experts agreed with scores ≥3 points (5-point rating while 5 referred to high relevance) on the scale. The suggestions of the experts were considered in modifying the scale, if any. Six experts evaluated the content of the K-FS-8 and the overall CVI calculated for K-FS-8 was 80%. The panel confirmed that no further modification was needed for the K-FS-8. Additionally, cognitive debriefing interviews were conducted with 7 general Korean adults who did not involve in this study as a participant for face validity. The scale was then administered to 7 adults with different sociodemographic characteristics whose education level were primary school in order to assess the face validity, clarity and readability of the translated items [28]. All of the 7 adults stated that the scale was understandable, and no further change was required. Then, the final version of K-FS-8 was pretested in monolingual (Korean) populations. The Korean version of fear scale(K-FS-8) was confirmed and available in the Supplementary file 1."

- Haynes, S. N., Richard, D. C. S., & Kubany, E. S. (1995). Content validity in psychological assessment: A functional approach to concepts and methods. Psychological Assessment, 7(3), 238–247. https://doi.org/10.1037/1040-3590.7.3.238

R1-6. I understand this was a survey, but based on research question, did the authors have any estimation for sample size prior to conducting the study?

RESPONSE: Thank you for the thoughtful comment. As this was a psychometric evaluation study. We did think about the sample size estimation. Prior to the commencement of the study, the factor analysis was proposed in our study protocol and therefore we considered the guideline for sample size estimation of conducting factor analysis. According to Mundfrom and colleagues’ recommendation for conducting factor analysis (2009), the suggested minimums for sample size include from 3 to 20 times the number of variables and absolute ranges from 100 to over 1,000. In this study, K-FS-8 includes 8 items and 160 participants (8 times 20) will be considered appropriate to conduct factor analysis. Therefore, our approach using a population-based cross-sectional online survey has reached 2000+ participants which we are confident to report that the sample size is sufficient.

- Mundfrom, D.J., Shaw, D.G. and Ke, T.L., 2005. Minimum sample size recommendations for conducting factor analyses. International Journal of Testing, 5(2), pp.159-168, DOI: 10.1207/s15327574ijt0502_4

R1-7. The authors state that PHQ-4 and PC-PTSD-5 were adopted for this study to assess convergent validity. What do the authors mean with this statement? Were these instruments not already validated in the Korean population?

RESPONSE: Thank you. We added more details on this regarding the convergent validity in 2.3.2. Convergent validity, known-group validity, internal consistency and reliability of the K-FS-8 section. Both PHQ-4 and PC-PTSD-5 were used to evaluate the convergent validity of the K-FS-8. Convergent validity indicates how closely a test is related to other tests that measure similar constructs. The PHQ-4 and PC-PTSD-5 share similar constructs with the fear scale and were widely adopted in public general. Therefore, these instruments were used for assessing the convergent validity of the scale (K-FS-8).

R1-8. For known- group validity, T test and ANOVA have been used. I wonder if the authors checked for the distribution of the data before going with the tests. Please clarify.

RESPONSE: Yes, we have examined the normal distribution of all variables using P-P plots. As we have included many figures in the Supplementary file 2, we decided not to supplement the P-P plots. Also, according to the Central Limit Theorem (CLT), sample sizes equal to or greater than 30 are often considered sufficient for the CLT to hold that the distribution of sample means approximates a normal distribution. Therefore, the t-test and f-test used in this study did not violate its assumptions. 

R1-9. Item response theory (IRT)- were the assumptions for IRT met?

RESPONSE: Thank you for the comment. The assumptions of IRT were met and relevant analyses were reported for (1) unidimensionality by EFA and parallel analysis; (2) local independence by item characteristics and item fit in Table 5.

R1-10. Results: Were there any missing data on any items? Please clarify.

RESPONSE: We have clarified the missing data in the manuscript as follows:

“Participants spent approximately 15 min to complete the survey (n.b., participants were required to answer all survey questions to complete the survey, hence no missing data).”

R1-11. Discussion: The discussion would benefit by including more literature in relation to fear scales in the local and the global context and the limitations with the existing scales.

RESPONSE: We have added more literature review as follows: 

“Apart from the BCFS for COVID-19 fear, DK Ahorsu, CY Lin, V Imani, M Saffari, MD Griffiths and AH Pakpour [42] also developed the Fear of COVID-19 scale(FCV-19; 7 items), which has been validated in several countries including South Korea. Those studies validated the FCV-19 by measuring the Cronbach’s α coefficient which ranged from 0.80 to 0.88[23, 43-47].”

R1-12. The authors mention that fear might be higher in Korean sample. What do the authors mean? Also, the fact that fear is a natural response to any threat, perceived or real and is protective. So, high level of fear during covid-19 at a time when nothing was known about it is justified. I wonder why the authors focus on the statement that high level of fear means it always needs intervention. As also reported in the conclusion. I believe the discussion should focus more on clinical or morbid fear that necessitates intervention, by doing a more through investigation. I suggest the authors consider this in discussion as well as conclusion. The authors have pointed out the limitations with the study, which is good.

RESPONSE: Thank you for your comment. We have revised the part as follows: 

“As previously mentioned, while fear can aid compliance to public health measures in a pandemic, higher levels of fear can precipitate increased stress levels and anxiety, triggering or exacerbating adverse health behaviours such as smoking [48] and excessive alcohol use [49]. Fear of COVID-19 can not only trigger domestic violence as a stress response, but can prevent victims from seeking help or medical attention for fear of contracting COVID-19 at hospitals [50]. Hence, there is greater significance in screening for people with higher fear levels due to the greater risk of them perpetuating adverse social and health behaviours, warranting psychological intervention as a means of primary and secondary prevention. Examples of effective psychological intervention include techniques in improving appraisal of the body, emotion regulation and attachment security, and adopting acceptance [51]. In this study, prevalence of COVID-19 fear in South Korea (84.6%) is almost twice to four times more than the prevalence of fear of COVID-19 in over 13 other countries (18.1% to 45.2%) in a scoping review study [52]. This indicates that even after stratification into levels of fear-related risk, there would be a greater number of South Korean people with clinically-significant fear requiring intervention, validating the need for proactive screening to identify them.”

R1-13. A critical aspect is that even if the correlations are high and indicate reliability, they might not be a valid indicator of what we intend to measure. So, this is also a limitation to studies which focus on one aspect of validity. The authors can consider this point also to highlight in their discussion. Other aspects such as divergent validity could also be discussed.

RESPONSE: Thank you for the constructive comment. In consideration of your comment, we have add a limitation in Discussion section as follows: 

“Although K-FS-8 showed a higher internal consistency of Cronbach’s α coefficient than existing COVID-19 fear scales, the finding should be interpreted with caution as additional reliability tests such as test-retest reliability was not performed”

Reviewer #2: 

R2-1. The manuscript is excellent in terms of the factor analysis and model fit calculations. 

RESPONSE: Thank you very much for your constructive comments. Please find our responses to your comments as follows.

R2-2. However, there are concerns about the theoretical foundations that require to be ironed out. The study attempts to validate a fear scale based on a non-communicable disease (breast cancer) for use in a context of a communicable disease (COVID-19), while the statistical analysis is sound, it is recommended that the authors strengthen their justification by communicating with the original authors of the BFCS to discuss the interchangeable nature of the scale, or provide and describe examples from the literature where this has been previously accomplished. More comments included in the attached document.

RESPONSE: We have contacted the original authors of the BFCS. We discussed about the use of BFCS for COVID-19 fear with the corresponding author, and the author approved the use of the BFCS for the validation to measure COVID-19 fear. In addition, the BFCS was successfully validated for measuring COVID-19 fear in Chinese population (Choi et al 2022). 

- Choi EPH, Duan W, Fong DYT, Lok KYW, Ho M, Wong JYH, Lin CC: Psychometric Evaluation of a Fear of COVID-19 Scale in China: Cross-sectional Study. JMIR Form Res 2022, 6(3):e31992.

We have added the validation information in Introduction as follows:

“Even though the fear scale was originally developed to measure the fear of breast cancer, a study validated the scale to measure the fear of COVID-19 using BCFS in mainland China and Hong Kong[22].”

R2-3. Please discuss existing scales (or related scales) in the native social context, and outline their shortcomings to further establish the significance of this study, it is recommended to provide stronger justification to use a scale from a different sphere (breast cancer). 

RESPONSE: Thank you for your comment. We have added more justification in Introduction as follows: 

“This study adopted the Breast Cancer Fear Scale (BCFS; 8 items) [19] for the psychometric evaluation. The BCFS was developed on the robust theoretical frameworks – Extended Parallel Process Model (EPPM)[20] and Health Belief Model (HBM)[21], which establishes the definitions of fear, threat and barrier based on a profound understanding of their nature[19]. Even though the fear scale was originally developed to measure the fear of breast cancer, a study validated the scale to measure the fear of COVID-19 using BCFS in mainland China and Hong Kong[22]. Moreover, the theoretical framework of BCFS (i.e., EPPM and HBM) has been widely used for research addressing public health emergencies including COVID-19. While several Korean validated COVID-19 fear scales are available [23, 24], the scales were neither validated with a larger Korean population, nor grounded in robust theoretical frameworks.”

R2-4. Furthermore, it is not clear how this scale will benefit in applied settings. 

Is it intended as a diagnostic measure? In the conclusion it was mentioned that healthcare professionals should use the K-FS-8 as a screening tool, can you elaborate on this point or provide examples from other countries where fear scales were used effectively in clinical settings to achieve favorable outcomes?

RESPONSE: We have revised the implication in Discussion part as follows: 

“)[22]. As previously mentioned, while fear can aid compliance to public health measures in a pandemic, higher levels of fear can precipitate increased stress levels and anxiety, triggering or exacerbating adverse health behaviours such as smoking [48] and excessive alcohol use [49]. Fear of COVID-19 can not only trigger domestic violence as a stress response, but can prevent victims from seeking help or medical attention for fear of contracting COVID-19 at hospitals [50]. Hence, there is greater significance in screening for people with higher fear levels due to the greater risk of them perpetuating adverse social and health behaviours, warranting psychological intervention as a means of primary and secondary prevention. Examples of effective psychological intervention include techniques in improving appraisal of the body, emotion regulation and attachment security, and adopting acceptance [51]. In this study, prevalence of COVID-19 fear in South Korea (84.6%) is almost twice to four times more than the prevalence of fear of COVID-19 in over 13 other countries (18.1% to 45.2%) in a scoping review study [52]. This indicates that even after stratification into levels of fear-related risk, there would be a greater number of South Korean people with clinically-significant fear requiring intervention, validating the need for proactive screening to identify them.”

R2-5. Breast cancer and COVID-19 are two fundamentally different health hazards, which may raise concerns if the same instrument is used to measure fear towards each. For example, breast cancer involves a hereditary risk that may not be shared by a pandemic disease. Similarly, a pandemic disease like COVID-19 spreads through contagion while breast cancer does not. Therefore, the fear towards each may manifest differently. It is not clear if the shared theoretical framework (EPPM/HBM) involve an overlap between non-communicable diseases and communicable diseases. These points do not undermine the current study, but the authors are strongly recommended to elaborate on their justification. It is recommended to get in touch with the original authors of the BFCS, and discuss these theoretical foundations that may prevent or facilitate the use of BFCS towards a communicable health hazard like COVID-19. Providing a citation to the correspondence with the BFCS authors’ approval will bolster this study’s theoretical foundations, especially considering that only one word was changed to convert the BFCS into a COVID fear scale.

RESPONSE: Thank you for your thoughtful comment. Please refers to our response to R2-2. 

As per EPPM and HBM, it is well documented in existing literature about the use of theoretical frameworks for the public health emergency including communicable diseases such as Yahaghi et al (2021) and Tesema et al (2021) – HBM; Yoon et al (2022) and Barnett et al (2014) – EPPM.

- Yahaghi, R., Ahmadizade, S., Fotuhi, R., Taherkhani, E., Ranjbaran, M., Buchali, Z., Jafari, R., Zamani, N., Shahbazkhania, A., Simiari, H. and Rahmani, J., 2021. Fear of COVID-19 and perceived COVID-19 infectability supplement theory of planned behavior to explain Iranians’ intention to get COVID-19 vaccinated. Vaccines, 9(7), p.684.

- Tesema, A.K., Shitu, K., Adugna, A. and Handebo, S., 2021. Psychological impact of COVID-19 and contributing factors of students’ preventive behavior based on HBM in Gondar, Ethiopia. PLoS One, 16(10), p.e0258642.

- Yoon, H., You, M. and Shon, C., 2022. An application of the extended parallel process model to protective behaviors against COVID-19 in South Korea. Plos one, 17(3), p.e0261132.

- Barnett, D.J., Thompson, C.B., Semon, N.L., Errett, N.A., Harrison, K.L., Anderson, M.K., Ferrell, J.L., Freiheit, J.M., Hudson, R., McKee, M. and Mejia-Echeverry, A., 2014. EPPM and willingness to respond: the role of risk and efficacy communication in strengthening public health emergency response systems. Health communication, 29(6), pp.598-609.

In consideration of your comment, we have added more explaination in Intoruction as follows: 

“Moreover, the theoretical framework of BCFS (i.e., EPPM and HBM) has been widely used for research addressing public health emergencies including COVID-19.”

R2-6. Please provide examples from the literature on what kind of interventions are feasible in this situation.

RESPONSE: We have added the examples in Discussion section as follows: 

“Hence, there is greater significance in screening for people with higher fear levels due to the greater risk of them perpetuating adverse social and health behaviours, warranting psychological intervention as a means of primary and secondary prevention. Examples of effective psychological intervention include techniques in improving appraisal of the body, emotion regulation and attachment security, and adopting acceptance [51].”

Reviewer #3

R3-1. Dear authors, this paper is interesting. However, it aims to explore the Psychometric Evaluation of the Korean version of the COVID-19 Fear Scale (K-FS-8). Before accepting this paper, significant corrections should be applied.

RESPONSE: Thank you very much for your constructive comments. Please find our responses to your comments as follows.

Abstract:

R3-2. Online survey by which program.

RESPONSE: We conducted the survey through a survey company using the company’s online survey platform. Due to word limitation in Abstract, we have explained the platform in the main text (2.3.1. Sample for the psychometric testing)

R3-3. PTSD and IRT scales which language that you use?

RESPONSE: All scales we used were validated in Korean. We have explained the Korean validation in the main text (2.3.2. Convergent validity, known-group validity, internal consistency and reliability of the K-FS-8)

R3-4. Only translation and convergent validity were used. Please add KMO test and Bartlett test of sphericity; you measure only Cronbach's alpha for reliability. please calculate compost also; Add total variance to your findings.

RESPONSE: Thank you very much for your comments. Due to word limitation in Abstract, we have explained all the results in the main text (3.4. Item response theory (IRT)-based analysis).

R3-5. Introduction: Nice introduction, but you mention the aim of your study at the end. Any related scales (fear) globally were not published before. Write down several validated scales to measure phycological concepts before your aims and then give your readers why this scale is important among the Korean population.

RESPONSE: We have provided the aim followed by further explanation for the aim at the end of Introduction. We have also introduced other available fear scales and justified the differences between our fear scale and the others as follows: 

“However, fear of COVID-19 in South Korea has not been investigated using a large sample. Therefore, this study aimed to measure the COVID-19 related fear with a large sample of Korean population during COVID-19 pandemic, by conducting a psychometric evaluation of the Korean version of fear scale. This study adopted the Breast Cancer Fear Scale (BCFS; 8 items) [19] for the psychometric evaluation. The BCFS was developed on the robust theoretical frameworks – Extended Parallel Process Model (EPPM)[20] and Health Belief Model (HBM)[21], which establishes the definitions of fear, threat and barrier based on a profound understanding of their nature[19]. Even though the fear scale was originally developed to measure the fear of breast cancer, a study validated the scale to measure the fear of COVID-19 using BCFS in mainland China and Hong Kong[22]. Moreover, the theoretical framework of BCFS (i.e., EPPM and HBM) has been widely used for research addressing public health emergencies including COVID-19. While several Korean validated COVID-19 fear scales are available [23, 24], the scales were neither validated with a larger Korean population, nor grounded in robust theoretical frameworks.”

More comparisons between the two are available in Discussion section. 

R3-6. Please check your references (Endnote) Champion, CS Skinner, U Menon, S Rawl, RB Giesler, P Monahan, and J Daggy

RESPONSE: We have revised the reference. 

Methods

R3-7. COSMIN Study Design what is this? More Explanation

RESPONSE: Thank you for your comment. The COSMIN Study Design checklist is recommended for designing studies to evaluate measurement properties of existing participant reported outcome measures. It can be used by researchers who are designing a study to evaluate measurement properties of an existing participant reported outcome measures. We have provided the reference to the sentence (i.e., Reference number 25).

R3-8. A 6 expert what are the results, and what about the Important Score? Did they make any modifications to fear items? It is not clear

RESPONSE: We have added the result of the panel review as follows: 

“The content relevance, clarity, and the accuracy of translation of the scale were considered to calculate the content validity index (CVI), of which a value ≥80% was used as the assessment standard, the proportion in which ≥80% of the experts agreed with scores ≥3 points (5-point rating while 5 referred to high relevance) on the scale. The suggestions of the experts were considered in modifying the scale, if any. Six experts evaluated the content of the K-FS-8 and the overall CVI calculated for K-FS-8 was 80%. The panel confirmed that no further modification was needed for the K-FS-8.”

R3-9. Inclusion criteria. above 20 years old. How you can detect that 15 years cannot participate or the same person can repeat the scale again and again? Also, did you add a consent form?

RESPONSE: We invited the panel members in a survey company as potential participants. The company registers verified members only, hence the repeated participation of the same participant can be prevented. Yes, the information about the consent form is available in 2.4. Ethical considerations section.

R3-10. Change sex to gender, please.

RESPONSE: We have changed ‘sex’ to ‘gender’ in the manuscript including Table 1. 

R3-11. Analysis: Please add KMO, and Bartlet test.

RESPONSE: We have provided the KMO and Barlett's test of sphericity in Table 4.

R3-12. Perfect analysis

RESPONSE: Thank you very much. 

R3-13. Discussion: Add what this study will add.

RESPONSE: We have revised the discussion section by adding the implications of the study findings.

R3-14. Compare what happened before COVID-19 and during it.

RESPONSE: Thank you for your comment. The contents about fear before COVID-19 were considered but with word limitations in the manuscript and as our research aimed to explore the public’s fear during the COVID-19 pandemic, we decided to keep our focus on fear only during that period.

R3-15. We do not prefer numbers on discussion; Add implications; References need critical revisions.

RESPONSE: We have revised the discussion section by adding the implications of the study findings. Moreover, we have reviewed and revised the references. 

Reviewer #4: Comments to the Author

R4-1. Authors have written a very clear, valuable, and extensive paper with significant number of subjects involved. I congratulate them on this piece of work. I have some minor comments for the authors to consider. It was very hard to mark parts of the manuscript because there were neither rows nor page numbers in the submitted manuscript. I will do my best to be as clear as possible.

RESPONSE: Thank you very much for your constructive comments. Please find our responses to your comments as follows.

R4-2. In the 2.3.1. Sample for the psychometric testing. You stated that “The eligibility criteria for participation were 1) adult (≥20 years old according to the civil law in South Korea) and 2) a South Korea resident who can read and understand the Korean language.” How can you be sure that participants in your study were Korean residents and that they were fluent in the Korean language? Please explain.

RESPONSE: One of the eligibility criteria in this study was the Koreans who live in South Korea. The survey company we used restricted the access of potential participants who live outside South Korea to the survey. Moreover, the survey was conducted in Korean. Therefore, their residency and Korean fluency can be warrantied.

R4-3. In the 2.3.2. Convergent validity, known-group validity, internal consistency and reliability of the K-FS-8. You stated that: “The test-retest reliability of PC-PTSD-5(r) was 0.83.” Was this result calculated in your study? if not, please add the reference.

RESPONSE: Thank you for your insightful comment. We have added the references as follows: 

“The test-retest reliability of PC-PTSD-5(r) was 0.83[30].”

R4-4. In the 3.1. Participants’ characteristics. You stated that response rate was 20.78%.

Is this an acceptable percentage for studies like this? Please provide reference to confirm that this percentage is acceptable. In addition, please, discuss how this low percentage of response could influence your results in limitation section.

RESPONSE: Thank you for your constructive comment. Previous review papers reported that response rate of online survey in the public research was more than 40%. However, increasing studies have reported that COVID-19 has negatively affected the survey response rates. While many online survey studies reported the similar response rates (i.e., around 20%), we acknowledge that the response rate in our study is low. In consideration of your comment, we have added the low response rate as a limitation of this study in Discussion section as follows:

“The low response rate in this study (i.e., 20.78%) might not represent the target survey population, which would cause potential nonresponse bias.”

R4-5. In the same section as above, you stated that 65.5% of the study participants had a Bachelor’s degree or above. What is the average educational level in Korea? Is sample in this study representative of the Korean population? If not, please discuss how high percentage of highly educated subjects could influence your results.

RESPONSE: Tertiary education attainment in South Korea is the highest among OECD countries (OECD, 2023). The average tertiary educational attainments between 25 and 64-year-olds and 25 and 34-year-olds were 51.7% and 69.3% in 2021 respectively (OECD, 2023). Therefore, the educational attainments between our study and the OECD report would be comparable. However, we understand your concern, thus we have added the limitation of this study in Discussion section as follows:

“The participants’ higher tertiary education attainment (69.3%) was slightly higher than the Korean general population (51.7% in an age group of 25 and 64-year-olds) [53]. The skewed distribution of the education level may potentially influence generalizability of the results. 

- OECD (2023), Population with tertiary education (indicator). doi: 10.1787/0b8f90e9-en (Accessed on 17 January 2023)

R4-6. Please revise the percentage calculation in Table 1 (socio-demographic profile) for education and the number of people living together.

RESPONSE: Thank you for your comment. We have revised the Table 1.

---

## [Editor Report · Decision Letter 1]

20 Feb 2023

Psychometric Evaluation of Korean version of COVID-19 Fear Scale (K-FS-8): a population based cross-sectional study

PONE-D-22-30798R1

Dear Dr. Fong,

We’re pleased to inform you that your manuscript has been judged scientifically suitable for publication and will be formally accepted for publication once it meets all outstanding technical requirements.

Kind regards,

Bimala Panthee

Academic Editor

PLOS ONE
---

## [Editor Report · Acceptance letter]

27 Feb 2023

PONE-D-22-30798R1 

Psychometric Evaluation of Korean version of COVID-19 Fear Scale (K-FS-8): a population based cross-sectional study. 

Dear Dr. Fong:

I'm pleased to inform you that your manuscript has been deemed suitable for publication in PLOS ONE. Congratulations! Your manuscript is now with our production department. 

Kind regards, 

on behalf of

Dr. Bimala Panthee 

Academic Editor

PLOS ONE